# Effects of Withdrawal Rate on the Microstructure of Directionally Solidified GH4720Li Superalloys

**DOI:** 10.3390/ma12050771

**Published:** 2019-03-06

**Authors:** Jinglong Qu, Shufeng Yang, Zhengyang Chen, Jingshe Li, Anping Dong, Yu Gu

**Affiliations:** 1Beijing Central Iron and Steel Research Institute, Beijing 100081, China; 13810256459@139.com (J.Q.); 18810654973@163.com (Y.G.); 2School of Metallurgical and Ecological Engineering, University of Science and Technology Beijing, Beijing 100083, China; chenzhengyang@xs.ustb.edu.cn (Z.C.); lijingshe@ustb.edu.cn (J.L.); 3Beijing Key Laboratory of Special Melting and Preparation of High-End Metal Materials, Beijing 100083, China; 4School of Materials Science and Engineering, Shanghai Jiao Tong University, Shanghai 200240, China

**Keywords:** withdrawal rate, GH4720Li, microstructure, average secondary dendrite arm spacing, γ’ phase, segregation coefficient

## Abstract

Increasing the ingot size of GH4720Li superalloys makes it difficult to control their microstructure, and the withdrawal rate is an important factor in controlling and refining the microstructure of GH4720Li superalloys. In this study, GH4720Li superalloy samples were prepared via Bridgman-type directional solidification with different withdrawal rates. The morphology and average size of the dendrites in the stable growth zone during directional solidification in each sample, morphology and average size of the γ’ phases, and microsegregation of each alloying element were analyzed using optical microscopy, Photoshop, Image Pro Plus, field emission scanning electron microscopy, and electron probe microanalysis. Increasing the withdrawal rate significantly helped in refining the superalloy microstructure; the average secondary dendrite arm spacing decreased from 133 to 79 µm, whereas the average sizes of the γ’ phases in the dendrite arms and the interdendritic regions decreased from 1.02 and 2.15 µm to 0.69 and 1.26 µm, respectively. Moreover, the γ’ phase distribution became more uniform. The microsegregation of Al, Ti, Cr, and Co decreased with the increase in the withdrawal rate; the segregation coefficients of Al, Cr, and Co approached 1 at higher withdrawal rates, whereas that of Ti remained above 2.2 at all the withdrawal rates.

## 1. Introduction

Superalloys exhibit outstanding high-temperature strength and resistance to oxidation, fatigue, and creep [1,2,3] and are key materials in the construction of high-temperature structural components in the aerospace sector. In particular, the GH4720Li superalloy is a new nickel-based age-hardened superalloy derived from the Udimet 720 superalloy [4,5]. This superalloy is being widely used in fabricating compressor discs, turbine discs, and turbine blades that operate at temperatures between 650 °C and 750 °C [6,7,8].

The rapid development in the aerospace sector and ground-based gas turbines has led to a progressive increase in the size of GH4720Li ingots. However, as GH4720Li superalloys are highly alloyed, increasing the ingot size makes it difficult to control their microstructure; this poses a significant threat to the safety and reliability of the superalloy [9,10]. To address this issue, some researchers have studied and manipulated the evolution of the strengthening γ’ phase in the microstructure to enhance the performance of these alloys. The metallurgical data regarding the GH4720Li superalloy show that different types of γ’ phases are formed during continuous cooling [11]. These γ’ phases can be classified into three types based on their size and distribution [12,13,14]. In the first type, the primary γ’ phase is quite large and is mainly observed in the grain boundaries. In the second type, the secondary and tertiary γ’ phases occur in the intragrain region, with the secondary γ’ phase being larger than the tertiary γ’ phase. Moreover, these phases exhibit a bimodal distribution. In the third type, the age hardening of the superalloy leads to solvation and growth of the tertiary γ’ phase, but does not significantly change the primary and secondary γ’ phases. In addition, the growth and roughening of the γ’ phases are diffusion-limited and consistent with the Lifshitz–Slyozov–Wagner (LSW) model, and the grain growth activation energy of the γ’ phase is in the range of 250–265 J/mol [15,16,17]. Xiao et al. [18] found that the creep processes of the GH4720Li superalloy under various conditions are controlled by dislocation climb and cutting.

Some researchers manipulated the microstructure of GH4720Li superalloys by adjusting their metallurgical process to improve the metallurgical quality and properties of the resulting GH4720Li ingot. For example, Wang et al. [19] discovered that the use of low-frequency alternating current in conjunction with a static transverse magnetic field during electroslag remelting could effectively reduce the size of the molten droplets and increase the reaction area of the slag–metal interface, thus improving the microstructure and metallurgical quality of the superalloy. Shevchenko et al. [20] demonstrated that time-dependent variations and asymmetry in the electric arc are detrimental to the slag discharge from the melt pool and steady-state solidification during consumable-electrode vacuum arc remelting. Hence, the ingot quality can be improved by controlling the electric arc in a rational manner. Chen et al. [21] found that triple-melt processes produce superalloy ingots that are superior in quality to those produced by double-melt processes and can effectively improve the overall mechanical properties of the superalloy.

In summary, many researchers have studied and manipulated the strengthening γ’ phase of the GH4720Li superalloy and adjusted the metallurgical processes to improve the performance of this superalloy. However, reports on the effects of the withdrawal rate on the microstructure of the GH4720Li superalloy (in terms of the interdendritic spacing, size and distribution of the γ’ phase, and microsegregation of the alloying elements) are limited, making it difficult to formulate parameters for the melting and heating of this superalloy. To address this issue, a Bridgman-type directional solidification furnace was used to prepare sample rods in this study, with the cooling rate being manipulated by varying the withdrawal rate. The effects of the withdrawal rate on the microstructure of the GH4720Li superalloy were then analyzed by observing and recording the morphology and average spacing of the dendrites and γ’ phase, as well as the microsegregation of each element. Our findings provide a theoretical foundation for the selection of process parameters to enhance the microstructure of GH4720Li superalloys.

## 2. Experimental Method

### 2.1. Melting of the Superalloy

GH4720Li superalloy samples prepared via the vacuum induction melting (VIM, Consarc, Rancocas, NJ, USA) + electroslag remelting (ESR, ALD, Hanau, Germany) + vacuum arc remelting (VAR, ALD, Hanau, Germany) process were used as the raw material in this study. Table 1 lists the chemical composition of the resulting GH4720Li superalloy. Figure 1 shows the phase diagram of the GH4720Li superalloy. The superalloy was remelted using the zone melting and liquid metal cooling method in a Bridgman-type directional solidification furnace (HeBei HanDan XiYuan High Frequency Instrument Co., Ltd., Handan, China). Throughout the remelting process, a vacuum pressure was maintained in the range of 0.03–0.08 Pa, while the temperature of the molten metal in the crucible was maintained at 1380 ± 10 °C. Finally, five withdrawal rates were applied to prepare five directionally solidified sample rods, each having a size of Ф6.5 mm × 100 mm. Table 2 lists the details of our experimental scheme, wherein the relationship between the cooling rate and the withdrawal rate is shown as Equation (1) [22]. In this equation, *v*_c_ is the cooling rate (in K/min); *v* is the withdrawal rate (in mm/min); and G is the temperature gradient (in K/mm), the value of which is 3.83 K/mm.
(1)$$v_{c}=Gv$$

### 2.2. Preparation and Testing

The five Ф6.5 mm sample rods were chemically etched in the longitudinal section in a solution containing 150 g CuSO_4_, 500 mL HCl, and 35 mL H_2_SO_4_. This procedure was used to determine the stable growth zone during directional solidification in these samples. Two Ф4 mm metallographic samples were then taken from the stable growth zone of each sample rod. Figure 2 shows the stable growth zone and sample extraction position of a sample rod. The samples have different macroscopic microstructures in the directional solidification process, wherein the microstructures of the sample are mainly equiaxed grain during the beginning of directional solidification. As the withdrawal distance increases, the microstructures of the sample gradually change to columnar crystal and tend to grow stably. After the withdrawal distance reaches a particular threshold, the microstructures of the sample are complex equiaxed grain and columnar crystal.

An electron probe microanalyzer (EPMA, Shimadzu Corp., Kyoto, Japan) was used to observe the segregation of Al, Ti, Cr, and Co in the metallographic samples. The samples were then electropolished using a solution containing 20% H_2_SO_4_ and 80% CH_3_OH and etched using a solution containing 15 g CrO_3_, 10 mL H_2_SO_4_, and 150 mL H_3_PO_4_. The morphologies and average sizes of the γ’ phases in the dendrite arms and interdendritic regions of these samples were observed using a field emission scanning electron microscope (FE-SEM, Carl Zeiss AG, Jena, Germany).

Finally, a 5 g CuCl_2_ + 170 mL HCl + 10 mL H_2_SO_4_ solution was used to chemically etch the remaining samples. The morphologies and average sizes of the interdendritic spacing in these samples were observed and recorded using optical microscopy (OM, Leica, Wetzlar, Germany), Photoshop (Version CS6, Adobe Inc., Mountain View, CA, USA), and Image Pro Plus (Version 6.0, Media Cybernetics, Inc., Rockville, MD, USA). To ensure the accuracy of the interdendritic spacing measurements, each sample was divided into four zones; 30 different field-of-views (FOVs) were used to measure the interdendritic spacings of each zone.

## 3. Results and Discussions

### 3.1. Analysis of Interdendritic Spacing

Figure 3 shows the microstructures of the stable growth zone of the GH4720Li superalloy samples prepared at five different withdrawal rates. The frontal edge of the solidification interface grows in the form of dendrites in all the samples. As the withdrawal rate is increased from 0.18 to 2.4 mm/min, the morphologies of the dendrite arms are significantly refined, while the number is visibly increased. Moreover, the number of dendrites are also increased in the dendrite arm regions. As a result, the dendrites in the samples gradually become cross-shaped, and the spacing between the dendrite arms steadily decreases. This is because the cooling rate of the sample increases in proportion with the withdrawal rate, and the increase in the cooling rate increases the heat dissipation at the solidification interface, thus reducing the size of the area affected by the latent heat of solidification. Consequently, the dendrite morphology of the sample becomes significantly more refined [23]. The changes in the interdendritic spacing were investigated in further detail by analyzing the secondary dendrite arm spacing (SDAS) of the samples with respect to the withdrawal rate. Figure 4 shows the results of this analysis. The relationship between the SDAS and the withdrawal rate is nonlinear, fitted using the power function shown in Equation (2). In this equation, *λ*_2_ is the SDAS (in μm); *v* is the withdrawal rate (in mm/min); *G_L_* is an empirical coefficient, the value of which is 0.8078; and the value of correlation coefficient is 0.986.
(2)$$\lambda_{2}=87.42{\left(G_{L}v\right)}^{-0.2148}$$

Figure 4 shows that the SDAS decreases from 133 to 99 μm (a 26% decrease) with the increase in the withdrawal rate from 0.18 to 0.6 mm/min; when the withdrawal rate is increased from 0.6 to 2.4 mm/min, the SDAS decreases from 99 to 79 μm (a 19% decrease). It may be inferred that the SDAS decreases more slowly at higher withdrawal rates. This is because of the relatively slow grain nucleation and growth during near-equilibrium solidification; if the cooling rate of the sample is increased at this point (i.e., by increasing the withdrawal rate), the degree of supercooling and the driving force required for phase transformation will increase as well; this helps in increasing the number of grain nuclei in the alloy, thus reducing the average SDAS [24]. However, after the cooling rate reaches a particular threshold, the driving force required for phase transformation due to any further increase in the cooling rate will exceed the driving force required for grain nucleation and growth. Hence, the influence of the cooling rate on the SDAS will be weaker at this point, and the decrease in the SDAS with further increase in the cooling rate will be less pronounced.

### 3.2. Analysis of γ’ Phase

Figure 5 shows the microstructures of the γ’ phase in the GH4720Li superalloys formed at five different withdrawal rates. Table 3 lists the average sizes of the γ’ phase in the dendrite arms and interdendritic regions. Based on Figure 5 and Table 3, the γ’ phase particles are irregular in the dendrite arms (Figure 5a–e), with the difference between the highest and lowest average sizes being 0.33 μm (Table 3, columns three and four). In the interdendritic regions, the average size of the γ’ phase is significantly greater than that in the dendrite arms, and the morphology is much more cube-like. Furthermore, the maximum average size of the γ’ phase is 1.71 times the minimum average size in these regions. Upon further analysis, it can be observed that the average γ’ phase size and its variance and fluctuations generally decrease with the increase in the withdrawal rate. Hence, the γ’ phases in the dendrite arms and interdendritic regions become more uniform at higher withdrawal rates.

During the solidification of the GH4720Li superalloy, the γ solid solution precipitates first from the liquid phase, and the L → γ + γ’ eutectic reaction occurs toward the end of the solidification process. After the superalloy is completely solidified, the solvation of the solute atoms in the γ solid solution decreases with the decrease in the superalloy temperature, and a precipitation transformation occurs at this instant in the oversaturated γ solid solution. This causes the γ’ phase to precipitate from the solution and gradually grow in size [25]. Hence, the morphology and size of the γ’ phase depends on the supersaturation and supercooling of the γ solid solution, the diffusion coefficient of the solute atoms in the γ solid solution, and the critical nucleation energy required for γ’ phase precipitation. In summary, the cooling rate of the GH4720Li superalloys increases in proportion with the withdrawal rate, and this increase leads to additional supercooling during γ’ phase precipitation, thus increasing the number of nucleation points. Furthermore, increasing the cooling rate decreases the growth time of the γ’ phase, thus reducing the average size of the γ’ phases in the dendrite arms and interdendritic regions. The distribution of the γ’ phase becomes more uniform as well [26,27].

### 3.3. Analysis of Microsegregation

The segregation of Al, Ti, Cr, and Co from the GH4720Li superalloys prepared at five different withdrawal rates was observed and recorded. Figure 6 shows the results, wherein the segregation coefficient K is the ratio of the element content in the dendrite arms to that in the interdendritic regions. The segregation of Al, Ti, Cr, and Co decreases with the increase in the withdrawal rate. Cr and Co, which are negative segregation elements, tend to aggregate in the dendrite arms and have segregation coefficients lower than 1 [28]; the difference between the highest and lowest segregation coefficients is less than 0.2. Al and Ti, which are positive segregation elements, tend to aggregate in the interdendritic regions and have segregation coefficients greater than 1. However, the segregation coefficient of Al approaches 1 at high withdrawal rates, whereas the segregation coefficient of Ti remains above 2.2.

These observations may be attributed to the fact that the segregation in the alloying elements in GH4720Li superalloys is mainly due to the change in the solid-solution phase and solute diffusion. Increasing the withdrawal rate (and therefore the cooling rate) inhibits the solute atom diffusion in the solid phase, but not in the liquid phase. Consequently, the solute contents in the solid phase deviate from the equilibrium composition and concentrate in the residual liquid phase. Once the cooling rate exceeds a certain threshold, solute diffusion will be completely suppressed in the solid phase and partially inhibited in the liquid phase, and the liquid–solid equilibrium distribution of the solute will only exist in the small amounts of the liquid phase near the solidification interface. If the cooling rate is increased at this point, the solute content in the residual liquid phase will decrease, thus reducing the element segregation in the GH4720Li superalloy [29,30]. Hence, the segregations of Al, Ti, Cr, and Co are reduced at higher withdrawal rates. Besides, the segregation between dendrite arms and interdendritic regions can be reduced by optimizing the melting process, improving the purity of the ingots or performing appropriate annealing treatment on the ingots after each melting process.

## 4. Conclusions

Increasing the ingot size of GH4720Li superalloys makes it difficult to control their microstructure. The influence of the withdrawal rate on the microstructure of the GH4720Li superalloys had so far been unclear, making it difficult to formulate parameters for the melting and heating of this superalloy. Therefore, we prepared GH4720Li superalloy samples via Bridgman-type directional solidification with different withdrawal rates and analyzed the morphology and average size of the dendrites in the stable growth zone during directional solidification, morphology and average size of the γ’ phases, and microsegregation of each alloying element. This study provided a theoretical foundation for the selection of process parameters to enhance the microstructure of GH4720Li superalloys. The following conclusions were drawn from this study:
As the withdrawal rate increases, the dendrites in the GH4720Li superalloy gradually become cross-shaped, and the average SDAS decreases from 133 to 79 µm. The relationship between the average SDAS and the withdrawal rate can be fitted using a power function (λ_2_ = 87.42 (*G_L_v*)^−0.2148^).As the withdrawal rate increases from 0.18 to 2.4 mm/min, the average sizes of the γ’ phase in the dendrite arms and interdendritic regions of the GH4720Li superalloy decrease from 1.02 and 2.15 μm to 0.69 and 1.26 μm, respectively. Moreover, increasing the withdrawal rate decreases the variance and fluctuation in the average size of the γ’ phase particles, indicating that the γ’ phases in the dendrite arms and interdendritic regions gradually become more uniform at higher withdrawal rates.In GH4720Li superalloys, Al and Ti are positive segregation elements, whereas Cr and Co are negative segregation elements. Overall, the segregation of Al, Ti, Cr, and Co decreases with the increase in the withdrawal rate; at higher withdrawal rates, the segregation coefficients of Al, Cr, and Co approach 1. However, the segregation coefficient of Ti remains above 2.2 at all the withdrawal rates.

## Figures and Tables

**Figure 1 materials-12-00771-f001:**
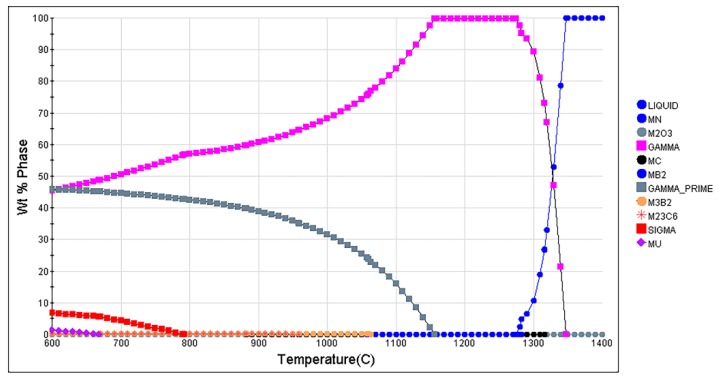
Phase diagram of GH4720Li superalloy.

**Figure 2 materials-12-00771-f002:**
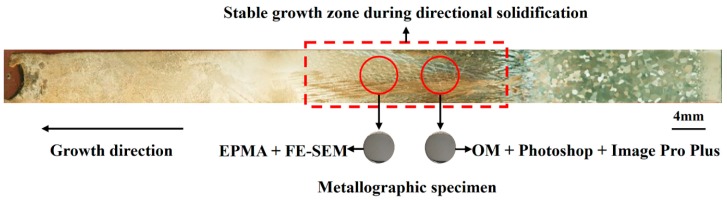
Schematic of the extracted metallographic sample.

**Figure 3 materials-12-00771-f003:**
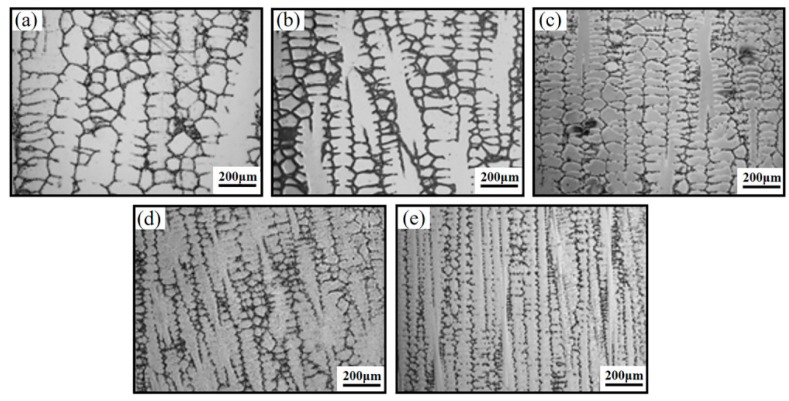
Microstructures of the stable growth zone of GH4720Li superalloys prepared at different withdrawal rates (mm/min): (**a**) 0.18; (**b**) 0.3; (**c**) 0.6; (**d**) 1.2; (**e**) 2.4.

**Figure 4 materials-12-00771-f004:**
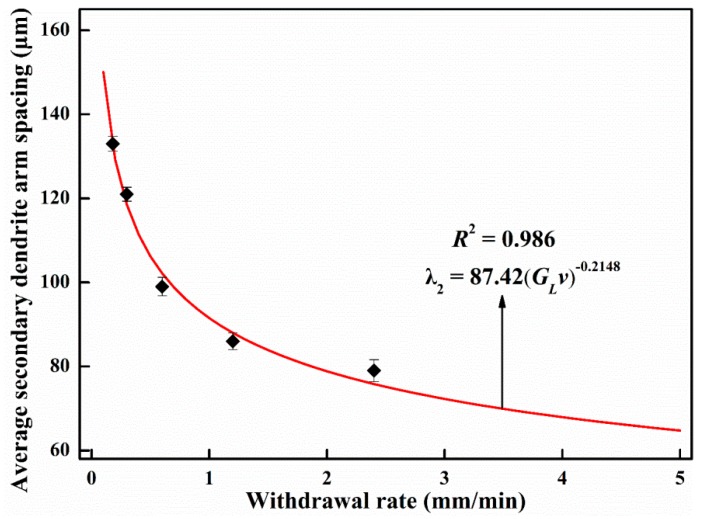
Average secondary dendrite arm spacing of GH4720Li superalloys with respect to the withdrawal rate.

**Figure 5 materials-12-00771-f005:**
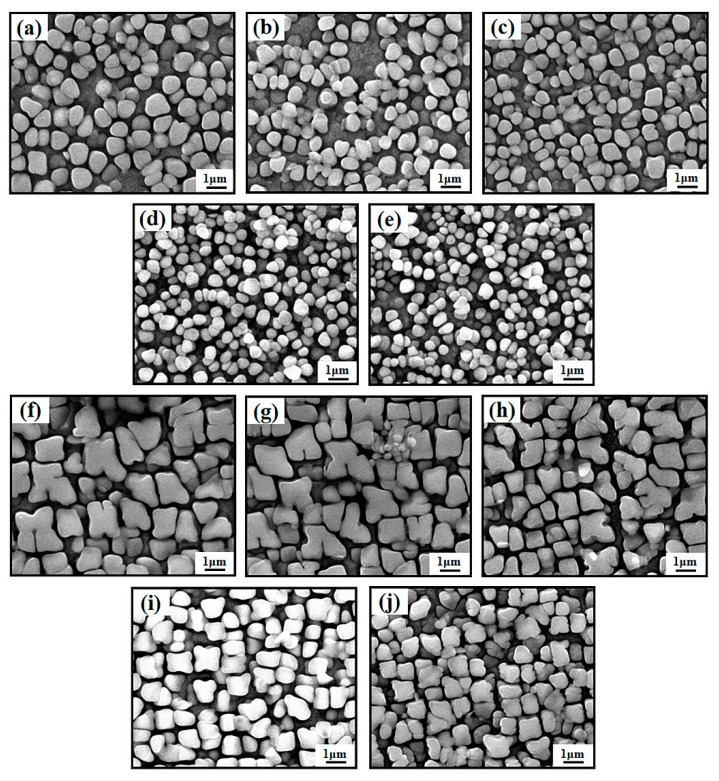
Structures of the γ’ phases in the dendrite arms (**a**–**e**) and interdendritic regions (**f**–**j**) of GH4720Li superalloy samples prepared at different withdrawal rates, from sample #1 to sample #5.

**Figure 6 materials-12-00771-f006:**
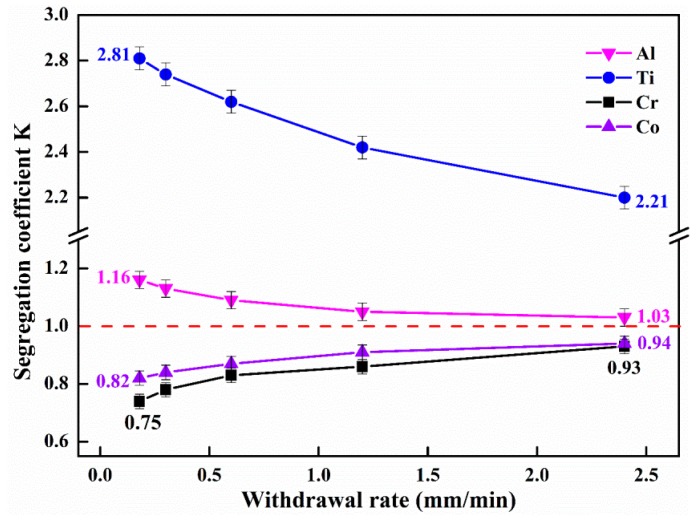
Segregation of elements in GH4720Li superalloy at different withdrawal rates.

**Table 1 materials-12-00771-t001:** Chemical composition of the GH4720Li superalloy (wt%).

Cr	Co	Ti	Al	W	B	S	Ni
15.85	14.75	4.99	2.53	1.26	0.015	≤0.015	Balance

**Table 2 materials-12-00771-t002:** Experimental scheme.

Sample No.	Withdrawal Rate (mm/min)	Cooling Rate (K/min)
#1	0.18	0.7
#2	0.3	1.2
#3	0.6	2.3
#4	1.2	4.6
#5	2.4	9.2

**Table 3 materials-12-00771-t003:** Average sizes of the γ’ phases in the dendrite arms and interdendritic regions of GH4720Li superalloy samples prepared at different withdrawal rates.

Sample No.	Withdrawal Rate (mm/min)	Average Size of γ’ Phase in the Dendrite Arm (μm)	Average size of γ’ Phase in the Interdendritic Region (μm)
#1	0.18	1.02 ± 0.15	2.15 ± 0.42
#2	0.3	0.94 ± 0.12	1.88 ± 0.36
#3	0.6	0.81 ± 0.08	1.61 ± 0.21
#4	1.2	0.74 ± 0.07	1.35 ± 0.15
#5	2.4	0.69 ± 0.05	1.26 ± 0.12

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
