# Peer review of "Effects of Withdrawal Rate on the Microstructure of Directionally Solidified GH4720Li Superalloys"

_materials, 2019, doi:10.3390/ma12050771_

Reviewer 1 Report

General comment: Please give some detailed information regarding GH4720Li alloy system regarding thermodynamic and kinetics, regarding γ’ phase stability etc. Perhaps an equilibrium isopheth phase diagram (optional), etc.

Line 21: Photoshop and Image Pro Plus are softwares for image processing not for elemental analysis such as SEM, TEM etc

firstly?

Please do not split up Fig. 8  in 2 pages.

Section 2.1: Melting of the superalloy

Where the cooling rate was measured? Weight of the 6.5 mm diameter rod and/or dimensions?

Section 2.2: Preparation and testing

the word ‘corroded’ (x3), was ‘corroded’ or ‘etched’?

Please enlarge Figures 1 and 2.

Equation 1 and Fig. 3: please give the correlation coefficient.

Line 156: …”between the highest and lowest average sizes being 0.33 μm”…please refer to values in 3rdand 4th columns in Table 3.

Figure 4: please identify the sample positions in Fig. 1 from where the samples were taken from.

Line 189: …”highest and lowest segregation”…

Please give values, what do you refer to?

Lines 210-212: “The influence of the withdrawal rate on the microstructure of the GH4720Li superalloys had far been unclear, making it difficult to formulate parameters for the melting, and heating of this superalloy”.

Author Response

Response to Reviewer 1 Comments

Point 1: General comment: Please give some detailed information regarding GH4720Li alloy system regarding thermodynamic and kinetics, regarding γ’ phase stability etc. Perhaps an equilibrium isopheth phase diagram (optional), etc.

Response 1:I would like to thank you for the careful and constructive reviews.The phase diagram of GH4720Li superalloy has been added in the manuscript. And the revised result is shown in Figure 1. The modified places have been marked using the ‘Track Changes’ function in Microsoft Word.Special thanks to your good comment. 

Figure 1. Phase diagram of GH4720Li superalloy. (in the manuscript)

Point 2: Photoshop and Image Pro Plus are softwares for image processing not for elemental analysis such as SEM, TEM etc. firstly?

Response 2:I would like to thank you for the careful and constructive reviews.In the manuscript, the electron probe microanalyzer (EPMA) is used to observe the segregation of Al, Ti, Cr, and Co, and the Photoshop and Image Pro Plus are used to observe the morphologies and average sizes of the interdendritic spacing. The modified places have been marked using the ‘Track Changes’ function in Microsoft Word.Special thanks to your good comment.

Point 3: Please do not split up Fig. 8 in 2 pages.

Response 3:I would like to thank you for the careful and constructive reviews.In this manuscript, there is no Fig. 8. And all the figures are not divided into two pages.Special thanks to your good comment.

Point 4: Section 2.1: Melting of the superalloy. Where the cooling rate was measured? Weight of the 6.5 mm diameter rod and/or dimensions?

Response 4:I would like to thank you for the careful and constructive reviews.The relationship between the cooling rate and the withdrawal rate (Equation (1)) has been added in the manuscript. In this equation, vc was the cooling rate (in K/min); v was the withdrawal rate (in mm/min); and G was the temperature gradient (in K/mm), the value of which was 3.83 K/mm. And the size of the directionally solidified sample rod was Ф6.5 mm × 100 mm. The modified places have been marked using the ‘Track Changes’ function in Microsoft Word.Special thanks to your good comment.                              (1) 

Point 5: Section 2.2: Preparation and testing. the word ‘corroded’ (x3), was ‘corroded’ or ‘etched’?

Response 5:I would like to thank you for the careful and constructive reviews.The word ‘corroded’ has been replaced with ‘etched’ in the manuscript. The modified place has been marked using the ‘Track Changes’ function in Microsoft Word.Special thanks to your good comment. 

Point 6: Please enlarge Figures 1 and 2.

Response 6:I would like to thank you for the careful and constructive reviews.Figures 1 and 2 have been enlarged in the manuscript. Due to the addition of the phase diagram of GH4720Li superalloy, the Figures 1 and 2 have been became Figures 2 and 3 respectively.Figure 2 (original Figure 1): The samples have different macroscopic microstructures in the directional solidification process, wherein the microstructures of the sample are mainly equiaxed grain during the beginning of directional solidification. As the withdrawal distance increases, the microstructures of the sample gradually change to columnar crystal and tend to grow stably. After the withdrawal distance reaches a particular threshold, the microstructures of the sample are complex equiaxed grain and columnar crystal.Figure 3 (original Figure 2): The frontal edge of the solidification interface grows in the form of dendrites in all the samples. As the withdrawal rate is increased from 0.18 to 2.4 mm/min, the morphologies of the dendrite arms are significantly refined, while the number is increased obviously. And the number of dendrites is also increased in the dendrite arm regions. As a result, the dendrites in the samples gradually become cross-shaped, and the spacing between the dendrite arms steadily decreases.The modified places have been marked using the ‘Track Changes’ function in Microsoft Word.Special thanks to your good comment. 

Point 7: Equation 1 and Fig. 3: please give the correlation coefficient.

Response 7:I would like to thank you for the careful and constructive reviews.The correlation coefficient in Equation 2 (original Equation 1) and Figure 4 (original Figure 3) has been added in the manuscript respectively. The modified places have been marked using the ‘Track Changes’ function in Microsoft Word.Equation 2 (original Equation 1): In Equation 2, λ2 was the SDAS (in μm); v was the withdrawal rate (in mm/min); GL was an empirical coefficient, the value of which was 0.8078, and the value of correlation coefficient was 0.986.Special thanks to your good comment.                                             (2) 

Figure 4. Average secondary dendrite arm spacing of GH4720Li superalloys with respect to the withdrawal rate. (in the manuscript)

 Point 8: Line 156: …“between the highest and lowest average sizes being 0.33 μm”

…please refer to values in 3rdand 4th columns in Table 3.

Response 8:I would like to thank you for the careful and constructive reviews.In the manuscript, the reference (Table 3 in 3rd and 4rd columns) has been added in the sentence (“between the highest and lowest average sizes being 0.33 μm”). The modified places have been marked using the ‘Track Changes’ function in Microsoft Word.Special thanks to your good comment. 

Point 9: Figure 4: please identify the sample positions in Fig. 1 from where the samples were taken from.

Response 9:I would like to thank you for the careful and constructive reviews.The figure 2 (original Figure 1) has been revised in the manuscript. The modified places have been marked using the ‘Track Changes’ function in Microsoft Word.Special thanks to your good comment. 

Figure 2. Schematic of the extracted metallographic sample. (in the manuscript)

 Point 10: Line 189: …“highest and lowest segregation” …Please give values, what do you refer to?

Response 10:I would like to thank you for the careful and constructive reviews.The figure 6 (original Figure 5) has been revised in the manuscript. And the highest and lowest values of the segregation coefficient have been given in the Figure 6. The modified places have been marked using the ‘Track Changes’ function in Microsoft Word.Special thanks to your good comment.

Figure 6. Segregation of elements in GH4720Li superalloy at different withdrawal rates. (in the manuscript)

 Point 11: Lines 210-212: “The influence of the withdrawal rate on the microstructure of the GH4720Li superalloys had far been unclear, making it difficult to formulate parameters for the melting, and heating of this superalloy”.

Response 11:I would like to thank you for the careful and constructive reviews.The sentence has been revised in the manuscript. The modified places have been marked using the ‘Track Changes’ function in Microsoft Word.

Special thanks to your good comment.

Reviewer 2 Report

Review of materials-454794

The authors have written a decent paper on solidification rate effects on dendritic (D) and interdendritic (ID) segregation, and their microstructure, specially arm spacings. While the results and interpretation are not new results and same results can be predicted with common sense. However, the rate of withdrawal can be useful for researcher who wish to work on this alloy. That gives this paper merit. I suggest accepting this paper, however please comment on few following points:

1)      What is the volume fraction of D and ID regions at different withdrawal rates?

2)      Can the authors suggest how to remove D and ID segregation?

Author Response

Response to Reviewer 2 Comments

Point 1: The authors have written a decent paper on solidification rate effects on dendritic (D) and interdendritic (ID) segregation, and their microstructure, specially arm spacings. While the results and interpretation are not new results and same results can be predicted with common sense. However, the rate of withdrawal can be useful for researcher who wish to work on this alloy. That gives this paper merit. I suggest accepting this paper, however please comment on few following points: What is the volume fraction of dendritic and interdendritic regions at different withdrawal rates?

Response 1:

I would like to thank you for the careful and constructive reviews.

The average size and volume fraction of γ’ phase in the dendrite arms and interdendritic regions are listed in Table 1. The results show the average size of γ’ phase and its volume fraction generally decrease with the increase of the withdrawal rate. The change ratio of the average size of γ′ phase in dendrite arms and interdentritc regions is 32.4% and 41.4% respectively, while the change ratio of the volume fraction is 9.0% and 8.6%, respectively. Obviously, the change ratio of the former is larger than the latter. This may be caused by the small variation range of the withdrawal rate designed in this study. In contrast, the average size of the γ' phase can more accurately characterize the effect of different withdrawal rates on the γ' phase in the GH4720li alloy. Therefore, the result of the volume fraction of γ' phase is not insert in the manuscript.

Special thanks to your good comment. 

Table 1. Average sizes and volume fraction of the γ' phases in the dendrite arms and interdendritic regions of GH4720Li superalloy samples prepared at different withdrawal rates.

Withdrawal rate

(mm/min)

Average size of γ′ phase (μm)

Volume fraction of γ′ phase (%)

dendrite

interdendritic region

dendrite

interdendritic region

0.18

1.02 ± 0.15

2.15 ± 0.42

47.8 ± 0.5

46.1 ± 0.6

0.3

0.94 ± 0.12

1.88 ± 0.36

46.5 ± 0.7

44.9 ± 0.8

0.6

0.81 ± 0.08

1.61 ± 0.21

45.3 ± 0.8

43.8 ± 0.5

1.2

0.74 ± 0.07

1.35 ± 0.15

44.4 ± 0.4

42.9 ± 0.8

2.4

0.69 ± 0.05

1.26 ± 0.12

43.5 ± 0.7

42.1 ± 0.7

Point 2: Can the authors suggest how to remove dendritic and interdendritic segregation?

Response 2:

I would like to thank you for the careful and constructive reviews.

The suggestions about removing dendritic and interdendritic segregation have been added in the manuscript. For example, the segregation between dendrite arms and interdendritic regions can be reduced by optimizing the melting process, improving the purity of the ingots or performing appropriate annealing treatment on the ingots after each melting process. The modified places have been marked using the ‘Track Changes’ function in Microsoft Word.

Special thanks to your good comment.
